Improving Gaussian Naive Bayes classification on imbalanced data through coordinate-based minority feature mining

Wang Wei
Yan Li yanli_gtu@sina.com
Liu Fen
Li Yanxi
School of Business, Guilin Tourism University , Guilin, Guangxi , China
Nakaya Helder
Electronic publication date: 2025 Jul 3
Publication date: 2025
Volume: 11
Electronic Location ID: e3003
Received 2025 Mar 7; Accepted 2025 Jun 11
Copyright: © 2025 Wang et al.
Copyright year: 2025
Copyright holder: Wang et al.
License: This is an open access article distributed under the terms of the Creative Commons Attribution License, which permits unrestricted use, distribution, reproduction and adaptation in any medium and for any purpose provided that it is properly attributed. For attribution, the original author(s), title, publication source (PeerJ Computer Science) and either DOI or URL of the article must be cited.
License URL: https://creativecommons.org/licenses/by/4.0/

Keywords: Coordinate transformation, Imbalanced data, Gaussian Naive Bayes classifier, Minority class salient features

Funding: Tourism Intelligent Financial Research Center Glky20200386 Guilin Tourism University The work was supported by the Tourism Intelligent Financial Research Center (Grant ID: Glky20200386) of Guilin Tourism University. The funders had no role in study design, data collection and analysis, decision to publish, or preparation of the manuscript.

==============================
As a widely used classification model, the Gaussian Naive Bayes (GNB) classifier experiences a significant decline in performance when handling imbalanced data. Most traditional approaches rely on sampling techniques; however, these methods alter the quantity and distribution of the original data and are prone to issues such as class overlap and overfitting, thus presenting clear limitations. This article proposes a coordinate transformation algorithm based on radial local relative density changes (RLDC). A key feature of this algorithm is that it preserves the original dataset’s quantity and distribution. Instead of modifying the data, it enhances classification performance by generating new features that more prominently represent minority classes. The algorithm transforms the dataset from absolute coordinates to RLDC-relative coordinates, revealing latent local relative density change features. Due to the imbalanced distribution, sparse feature space, and class overlap, minority class samples can exhibit distinct patterns in these transformed features. Based on these new features, the GNB classifier can increase the conditional probability of the minority class, thereby improving its classification performance on imbalanced datasets. To validate the effectiveness of the proposed algorithm, this study conducts comprehensive comparative experiments using the GNB classifier on 20 imbalanced datasets of varying scales, dimensions, and characteristics. The evaluation includes 10 oversampling algorithms, two undersampling algorithms, and two hybrid sampling algorithms. Experimental results show that the RLDC-based coordinate transformation algorithm ranks first in the average performance across three classification evaluation metrics. Compared to the average values of the comparison algorithms, it achieves improvements of 21.84%, 33.45%, and 54.63% across the three metrics, respectively. This algorithm offers a novel approach to addressing the imbalanced data problem in GNB classification and holds significant theoretical and practical value.

Introduction

In the era of big data, data is widely applied in various fields such as healthcare, finance, image recognition, and natural language processing. However, imbalanced data distribution is a common issue in datasets. Although minority class samples account for a small proportion, they often contain critical information. For example, in scenarios such as rare disease diagnosis and fraud transaction detection, accurately identifying minority class samples is of great significance.

The Gaussian Naive Bayes (GNB) classifier is based on Bayes’ theorem and the assumption of feature conditional independence. It is widely used in various classification tasks due to its simple principle, high computational efficiency, adaptability to small-scale data, and ability to handle multi-class problems (Kamel, Abdulah & Al-Tuwaijari, 2019). However, GNB classifier’s performance is severely affected when dealing with imbalanced data. Since GNB classifier performs probabilistic calculations and classification decisions based on idealized assumptions, the majority class dominates the probability calculation in imbalanced datasets. This leads to the model’s insufficient learning of minority class features, which results in low classification accuracy and a high likelihood of misclassification (Sağlam & Cengiz, 2024). Moreover, GNB classifier’s estimation of prior probabilities relies on sample size, and imbalanced distribution can bias this estimation. This, in turn, affects the posterior probabilities and classification results, reducing the model’s generalization ability and limiting its application in practical scenarios. Therefore, researching methods to improve the classification performance of GNB classifier on imbalanced data has important theoretical and practical significance.

To improve the performance of GNB classifier on imbalanced data, researchers have proposed improvement measures at the algorithm and data levels. At the algorithm level, the aim is to prevent the classifier from being biased towards the majority class. For example, improving the prior probability estimation and introducing a weighting strategy to make it more in line with reality (Carrizosa, Nogales-Gómez & Morales, 2021); through feature selection and weight adjustment, screening key features and reasonably allocating weights to make the model pay attention to the features of the minority class (Fu et al., 2022); adopting ensemble learning to combine multiple GNB classifiers to improve the robustness and classification ability of the model (Gayathri & Sumathi, 2016). However, algorithm improvement is difficult, highly complex, and requires a large amount of computing resources, which limits its practical application. Improvements at the data level are more common, mainly represented by sampling methods. The idea of sampling methods is to adjust the number of samples to achieve class balance, including undersampling, oversampling, and hybrid sampling (Wang et al., 2021).

Undersampling methods directly and randomly select a certain number of samples from the majority class samples for deletion, thereby reducing the proportion of the majority class samples and achieving the balance of the dataset. Elhassan et al. (2016) proposed the Tomek Links undersampling method, which identifies Tomek Links pairs in the dataset and then deletes the majority class samples among them. This reduces the number of majority class samples while retaining those samples near the classification boundary that are of great significance to the classification decision (Elhassan et al., 2016). Ma & Li (2019) proposed the AllKNN method, which uses the characteristics of the KNN algorithm to delete the majority class samples with low importance for classification, trying to retain the samples helpful for classification. This enables the dataset to reach a relatively balanced state, allowing the model to better learn the classification boundary and improve the classification ability for minority class samples (Ma & Li, 2019). The undersampling method is simple, straightforward, and easy to implement. However, achieving data balance by deleting majority class samples is likely to lose key information, affecting the model’s learning of the features of the majority class samples and reducing the overall classification performance and generalization ability.

Oversampling methods generate new samples based on the minority class samples. This can retain most of the information in the original data and help the model learn more comprehensive features, especially those of the minority class samples, so it has been more widely applied. A relatively classic oversampling method is the Synthetic Minority Over-sampling TEchnique (SMOTE) algorithm proposed by Chawla et al. (2002), which randomly selects one of the K nearest neighbors of the minority class samples and performs linear interpolation to synthesize new samples. The SMOTE algorithm can effectively improve the classification performance of imbalanced datasets, but it also has problems such as blurring the class boundaries and causing sample class overlap. Therefore, researchers have improved it from two aspects: clustering and boundary samples. Oversampling methods based on clustering mainly focus on intra-class balance. By clustering the minority class samples and then generating new samples within the clusters, the diversity of the minority class samples is ensured. Cieslak, Chawla & Striegel (2006) proposed the K-means SMOTE algorithm, which first clusters and then applies the SMOTE algorithm to each cluster separately. In view of the shortcomings of the K-means clustering method, Liu, Zhang & Zhao (2024) proposed the DPCSMOTE algorithm based on density peak clustering. Oversampling based on boundary samples focuses on inter-class balance. Commonly used algorithms include ADASYN (He et al., 2008), SVMSMOTE (Abdi & Hashemi, 2016), MWMOTE (Barua, Islam & Yao, 2014), Borderline-SMOTE (Han, Wang & Mao, 2005), etc. Yongxin et al. (2022) proposed the Local Density Borderline-SMOTE (LDBSMOTE) algorithm based on Borderline-SMOTE, considering the impact of the local density of the minority class on interpolation; Ma, Song & Zhu (2023) introduced a synthesis factor in the classification boundary division and proposed an Improved Borderline-SMOTE (IBSM) algorithm. The disadvantage of oversampling methods is that the model over-learns the minority class samples. It performs well on the training set but has poor generalization ability on the test set or new data and cannot accurately classify. At the same time, it may generate unreasonable samples or samples with too high similarity to existing samples, blurring the class boundaries and reducing the classification accuracy of the model.

In view of the shortcomings of undersampling and oversampling methods, some researchers have also proposed hybrid sampling methods that combine the two. This method first oversamples the minority class samples to relatively increase their proportion in the dataset; then, it undersamples the majority class samples to reduce their dominance in the dataset, so that the dataset reaches a more balanced state in terms of quantity while retaining more valuable information. For example, Batista, Prati & Monard (2004) proposed the SMOTETomek and SMOTEENN algorithms. However, hybrid sampling also has the problem of overprocessing the data, causing the dataset to lose some of its original features and distribution information. As a result, the knowledge learned by the model during the training process is not accurate enough, which instead reduces the generalization ability and classification performance of the model.

Overall, when addressing the data imbalance problem faced by the GNB classifier using sampling methods, two salient drawbacks are exposed. First, sampling methods inevitably alter the original data distribution and increase the risk of overfitting by adding or reducing the number of samples in the dataset to achieve class balance. Second, for imbalanced datasets with complex distributions, oversampling and hybrid sampling methods can lead to substantial class overlap, blurring class boundaries. On the other hand, undersampling often removes samples from boundary cases or special patterns, preventing the GNB classifier from learning the complete classification boundaries. These issues ultimately affect the classification performance of the GNB classifier.

To address these shortcomings, this article proposes a data preprocessing method based on radial local relative density changes (RLDC) coordinate transformation. The key feature of this method is that it does not change the original number or distribution of data. Instead, it transforms the data from the traditional absolute coordinate space to the RLDC relative position coordinate space, thereby deeply mining the local relative density variation features in the original data. These features are usually highly characteristic in minority class samples and can clearly reveal the unique patterns of minority class samples. In contrast, they are not as prominent in majority class samples. These new features provide the GNB classifier with more critical and highly discriminative information, effectively enhancing the conditional probability of minority class samples. As a result, the GNB classifier more can accurately determine whether a sample belongs to the minority class during the classification decision process. This not only saliently increases the recognition probability of minority class samples but also greatly improves the overall classification performance of the GNB classifier on imbalanced datasets.

Compared with locally weighted learning (LWL) (Sağlam & Cengiz, 2024; Frank, Hall & Pfahringer, 2012; Jiang et al., 2013; Wu et al., 2017), which also focuses on local data features, the RLDC coordinate transformation method proposed in this article demonstrates salient differences in core mechanisms and objectives: LWL is a dynamic non-parametric approach that performs local fitting through real-time calculation of weights for neighboring samples around query points. It is suitable for general regression or classification tasks, relying on distance relationships in the original feature space without altering data distribution, but exhibits insufficient sensitivity to data imbalance problems. In contrast, RLDC explicitly reconstructs the feature space through global coordinate transformation as a parametric method in the preprocessing stage, specifically designed to enhance classification performance on imbalanced data. While LWL incurs higher computational costs due to real-time weight calculations, RLDC achieves efficient classification during inference through distribution optimization in the training phase, with stronger interpretability (visualizable geometric transformations). This enables RLDC to more reliably supply effective feature information to GNB classifiers, thereby improving classification performance under imbalanced data conditions.

Materials and Methods

Principle of binary classification for GNB classifier

In the GNB classifier, for a binary classification problem, assume we have a class set C={Cma,Cmi}. Given a sample x={x1,x2,⋯,xn} within it, where xi represents the features of the sample. According to Bayes’ theorem, the probability that sample x belongs to class Ck is calculated as shown in Eq. (1) (Çınar, 2024).

(1) P(Ck|x)=P(x|Ck)P(Ck)P(x)

Since P(x) is the same for all classes, it can be ignored when comparing the probabilities of different classes. For the Gaussian Naive Bayes, assuming conditional independence among features, the formula for calculating the likelihood probability P(x|ci) is as follows.

(2) P(x|Ck)=∏i=1n12πσk,i2exp⁡(−(xi−μk,i)22σk,i2)

In Eq. (2), μk,i is the probability of the feature xi occurring under class Ck, and σk,i2 is the variance of the feature xi in class Ck.

The prior probability P(Ck) can usually be estimated by the proportion of samples of class Ck in the training data to the total number of samples, that is P(Ck)=|Ck||D|, where |Ck| is the number of samples of class Ck, and |D| is the total number of training samples.

The impact of minority class salient features on GNB classification performance

In real-world imbalanced data classification scenarios, “salient features of minority classes” refer to those characteristics that exhibit high distinctiveness and strong discriminative power in differentiating minority class samples from majority class samples. These features demonstrate unique distribution patterns or higher occurrence frequencies in minority class samples, while appearing less frequently or exhibiting different patterns in majority class samples. From an information content perspective, the salient features of minority classes contain rich information that captures the essential attributes of minority samples. Compared to other features, they play a more critical role in identifying minority class samples and serve as key distinguishing criteria between minority and majority classes. Regarding their importance, these salient features are decisive for the accurate classification of minority class samples. When effectively identified and leveraged, they can significantly enhance a classification model’s ability to recognize minority class samples in imbalanced data scenarios, thereby improving overall classification performance.

Due to data distribution imbalance, the limited quantity of minority class samples causes these features to manifest more concentrated and prominent patterns in the minority class, clearly reflecting their distinctive characteristics. In contrast, within the majority class, these features become diluted or obscured by the abundance of samples and relatively uniform distribution of other characteristics, losing their discriminative prominence. For instance, in diabetes screening scenarios, salient features of the minority class (diabetic patients) such as fasting plasma glucose (FPG) and glycated hemoglobin (HbA1c) maintain stable distributions in the majority class (healthy individuals) within normal ranges (e.g., FPG < 5.6 mmol/L), while showing salient clinical threshold breaches (e.g., FPG ≥ 7.0 mmol/L) or chronic abnormality accumulation (e.g., HbA1c ≥ 6.5%) in diabetic patients. Statistical analysis and model interpretation techniques (e.g., SHAP values) can validate the strong discriminative power of these features. When integrated with appropriate algorithms, they substantially improve diabetic patient identification accuracy while aligning with medical diagnostic standards.

In the complex scenario of imbalanced data classification, GNB always faces the challenge of improving performance. Particularly when dealing with minority class samples, the features of the majority class tend to dominate the data distribution, making the features of the minority class easily overshadowed and saliently reducing the classification effectiveness. However, when the features of the minority class are saliently prominent, the unique advantages of GNB based on the assumption of feature independence can be highlighted. Under this assumption, the conditional probabilities of these prominent minority class features (denoted as P(xi|Cmi) for the minority class Cmi) are usually at a relatively high level, and the recognition ability of GNB for minority class samples will also be greatly enhanced.

Taking fraud transaction detection in the financial field as an example, in a vast dataset of transactions, normal transactions account for the absolute majority, while fraud transactions belong to the minority class. When analyzing transaction data in depth, we can identify certain specific features, such as multiple large transactions in different locations within a short time and frequent changes in transaction IP addresses, which frequently occur in fraud transactions but are almost non-existent in normal transactions. According to the basic principle of estimating probability based on frequency, the conditional probability of these features under fraud transactions (minority class) is naturally high. When GNB makes classification decisions, based on the assumption of feature independence, these high-frequency features in fraud transactions will carry greater weight in the calculation of conditional probabilities. For example, assume that the conditional probability of feature xA (multiple large transactions in different locations within a short time) in fraud transactions is P(xA|Cmi), and the conditional probability of feature xB (frequent changes in transaction IP addresses) is P(xB|Cmi). Under the assumption of feature independence, the joint conditional probability of these two features appearing together in fraud transactions is P(xA,xB|Cmi)=P(xA|Cmi)×P(xB|Cmi). This calculation method effectively accumulates the weights of features related to fraud transactions, thereby saliently increasing the posterior probability of the fraud transaction category. When GNB encounters new transaction data, if it detects a large number of these high conditional probability features, it can more accurately classify it as a fraud transaction and effectively avoid misclassifying it as a normal transaction. This process fully demonstrates the key role that salient features of minority classes play in improving the classification performance of GNB in imbalanced data classification.

Meanwhile, in Bayesian theorem, the posterior probability P(Ck|x) also relates to the prior probability P(Ck). In imbalanced datasets, the minority class’s low prior probability P(Cmi) often leads to its neglect by Bayesian classifiers. However, when the minority class possesses salient features (e.g., “cross-region large-amount transfers” in fraudulent transactions or disease-specific symptoms in rare illnesses), their exceptionally high conditional probability P(x|Cmi) can significantly amplify the posterior probability P(Cmi|x) through the multiplicative effect of the Bayesian formula, thereby overriding the majority class’s dominance in probability calculations. For instance, when a symptom exhibits a conditional probability of 0.8 in a rare disease but only 0.01 in common diseases, even with the rare disease’s prior probability as low as 0.1%, its posterior probability can rise to 7.4%, triggering further diagnostic investigation. While this mechanism depends primarily on general class separation in balanced data, in imbalanced scenarios, the conditional probability must substantially exceed that of the majority class to offset prior probability disadvantages. This highlights how minority class salient features specifically address data imbalance challenges through targeted optimization, rather than relying solely on class separation improvements.

In summary, the strong correlation between minority class salient features and conditional probabilities provides critical insights for enhancing GNB’s classification performance in imbalanced data. However, in such scenarios, the majority class’s global distribution dominates the feature space, causing the minority class’s salient features to become obscured. Therefore, the RLDC coordinate transformation algorithm proposed in this article fundamentally addresses data imbalance by uncovering these masked salient features of minority class samples at the data distribution level. This enables them to play a more substantial role in GNB’s classification mechanism, thereby directly tackling data imbalance issues and comprehensively improving overall classification performance.

Mining ideas for salient features of minority classes

Under the premise of maintaining a constant sample size, the key to achieving dataset balance lies in identifying a unique observational perspective for analyzing samples. This new perspective enables the extraction of salient features from minority class samples while relatively diminishing the characteristics of majority class samples, thereby shifting the conditional probabilities of the two classes and ultimately balancing their posterior probabilities.

To effectively capture the spatial variation in local class proportions, this article introduces the concept of local relative density. local relative density is defined as the ratio of the number of minority class samples to the total number of samples within a radius neighborhood centered at a given sample point. In imbalanced datasets, salient differences exist in the local relative density variation characteristics between minority and majority classes. Majority class samples exhibit highly uniform and stable local relative density distributions, characterized by feature homogeneity and high information redundancy. This makes their numerical dominance ineffective in contributing meaningful information to classification boundaries. Minority class samples, however, display pronounced multimodal characteristics. Their local relative density fluctuates dramatically across the feature space, with abrupt transitions between sparse and dense regions forming a complex distribution pattern that carries high information entropy. Near classification boundaries, the density mutations of minority class samples often encode critical discriminative information. These distinct local relative density behaviors arise from three fundamental factors: • Differences in sample size and statistical stability

The majority class, with its large sample size, adheres to the law of large numbers, where local statistical estimates converge to the true distribution. For instance, normal transactions (majority class) consistently maintain a stable proportion of 90 ± 2% within the 100–500 transaction amount range, resulting in minimal fluctuations in local relative density even when new samples are added. In contrast, the sparsity of minority class samples leads to severe local statistical volatility. For example, fraudulent transactions (minority class) may account for only 0.1% of samples in the 5,000–10,000 transaction amount range, but within specific sub-ranges like 8,000–8,500, their local relative density can abruptly surge to 15% due to sporadic high-value transactions, forming sharp peaks in distribution.

• Distribution patterns and information structure divergence

Majority class distributions are typically simple (e.g., unimodal Gaussian) with strong feature homogeneity. For instance, normal user logins cluster around local IP addresses, exhibiting smooth majority proportion curves where local relative density gradually increases from the center to the periphery (3% → 5% → 10%). Minority classes, however, adopt complex multimodal distributions with salient feature heterogeneity. Fraudulent transactions may cluster in two distinct high-risk zones: Zone A (late-night microtransactions with 15% local relative density) and Zone B (cross-border macrotransactions with 20% local relative density), separated by low local relative density transition regions. This creates an alternating “sparse-dense” pattern.

• Classification boundary and information entropy contrast

Majority class samples predominantly lie far from classification boundaries, contributing minimal discriminative information due to low local entropy. In contrast, minority class samples cluster near decision boundaries, where high local entropy renders them critical for classification. For example, in academic performance evaluation, average students (majority) dominate low-score regions with limited informational value, while top performers (minority) near high-score boundaries carry decisive classification signals.

By characterizing imbalanced datasets through class-specific local relative density variations, this approach amplifies minority class features while simplifying majority class representations. This provides a robust and efficient methodology for dataset balancing.

Coordinate transformation algorithm based on radial local relative density changes

Based on local relative density, we can examine the two-class samples’ local density spatial variation by analyzing radial local relative density changes. The radial local relative density change is the ratio change of minority class sample number to the total two-class sample number within different radii around a center point along the radial direction. It describes the local density change of two-class sample distribution within a specific radial-distance range. In simple terms, it measures the density of the two-class sample in spheres with different radii centered at a certain point, emphasizing the radial distribution characteristics of the two-class sample, which helps to analyze the local aggregation or dispersion characteristics of the two types of sample data in space.

Following this idea, this article proposes a coordinate transformation algorithm based on radial local relative density changes (RLDC). From the perspective of each point in the space, this algorithm divides the entire data space into several radial intervals (i.e., different radius ranges). It calculates the ratio of the minority class sample count to the total count of the two classes within different radii (radial distances) with this point as the center. Then, it uses these ratios as the new features of this point to construct a new feature space. In this way, it can effectively capture the distribution characteristics of the two-class sample in the local area, enhancing the model’s ability to recognize the minority class.

In addition, in most cases, as the radius increases, the number of majority-class sample points in the neighborhood usually grows faster than that of the minority class, and the radial local relative density gradually decreases. The data distribution takes a form that is more consistent with the Gaussian function. This transformation provides strong support for improving the performance of the GNB classifier from another perspective. By making the data more in line with the characteristics of the Gaussian distribution, the GNB classifier can calculate the probabilities of features under various categories more accurately based on a data distribution that is more consistent with its assumptions when performing probability estimation and classification decision-making, thereby improving the accuracy and stability of classification.

Let the dataset Q contain two classes of samples, where the minority class sample subsets are D={xi1}, i=1,2,⋯,n, and the majority class sample subsets are F={xj2}, j=1,2,⋯,m. The distance between two points in dataset Q is calculated using the Euclidean distance, as shown in Eq. (3).

(3) dist(xa,xb)=∑lw(xal−xbl)2

In Eq. (3), w represents the number of dimensions of the data points. xal and xbl represent the values of data points xa and xb in the dimension l, respectively.

The local relative density ρa within a neighborhood centered at sample point xa with a truncation distance dc as the radius is calculated as shown in Eq. (4).

(4) ρa=numSnums+numg

In Eq. (4), nums is the number of minority class samples within the neighborhood, and numg is the number of majority class samples within the neighborhood.

For a given center point xa and a series of radii r1,r2,⋯,rn, the local relative density ρai within the neighborhood centered at xa with radius ri is calculated. The local relative densities under different radii are then integrated into a new feature vector xa′, as shown in Eq. (5).

(5) xa′=[ρ1a,ρ2a,⋯,ρna]

For each sample point xa in dataset Q, a corresponding new feature vector xa′ can be obtained through the above method. In this way, a new feature space Q′ is constructed, where each sample point xa is represented by its corresponding new feature vector xa′ in this new space Q′. In this new feature space, the relationships and distributions of sample points are described and analyzed based on these new feature vectors, thereby highlighting the characteristics of local relative density changes.

Algorithm steps.

Input: Initial sample set Q	
Output: Sample set Q′ after oversampling processing	
STEP1: Calculate the distances between all points in Q	
  1.1. Let Q be the set of points, containing m points.	
  1.2. Initialize a distance matrix D of size m × m.	
  1.3. For each point qi in Q:	
    1.3.1. For each point qj in Q (where j > i):	
      1.3.1.1. Calculate the distance dij between qi and qj.	
      1.3.1.2. Store dij in the distance matrix D at position (i, j).	
  1.4. Return the distance matrix D.	
STEP2: Calculate the radial local relative density at n thresholds based on the distance matrix	
  2.1. Let n be the number of thresholds.	
  2.2. Initialize a list of thresholds T, containing n distinct thresholds.	
  2.3. Initialize a new dataset Q′.	
  2.3. For each point qi in Q:	
    2.3.1. For each threshold tj:	
      2.3.1.1. Calculate the number of points mi, j within distance tj from qi.	
      2.3.1.2. Calculate the number of minority points ni, j within distance tj from qi	
      2.3.1.3. Calculate the local relative density ri, j for point qi at threshold tj as ri, j = ni, j/mi, j.	
    2.3.2. Add these local relative densities as new features for point qi in the new dataset Q′.	
  2.4. Return the new dataset Q′.	

Analysis of algorithm complexity

For a dataset with m samples and d-dimensional features, the algorithm involves two main steps:

The step of calculating pairwise distances between all points has a time complexity of O(m2⋅d). The step of computing local relative density for all points under n distance thresholds has a time complexity of O(m2⋅n). Thus, the total computational complexity is O(m2⋅d), exhibiting quadratic time complexity. This aligns with the quadratic complexity of common sampling algorithms like SMOTE and K-SMOTE, so the time complexity is acceptable. However, the space complexity for storing the full distance matrix grows as O(m2), becoming a critical bottleneck for medium-to-large-scale datasets.

To address these limitations, the following optimizations are implemented: KDTree hierarchical indexing replaces full pairwise distance computation with spatial partitioning, reducing the time complexity from O(m2⋅d) to O(mlog⁡m⋅d).

Dynamic radius-based neighbor retrieval utilizes radius queries (query_radius) to dynamically retrieve neighborhoods for each sample, avoiding precomputed matrices. This reduces the local relative density computation time under n thresholds from O(m2⋅n) to O(n⋅m⋅k), where k is the average number of neighbors per sample.

Sparse matrix storage retains only valid neighborhood relationships, compressing the space complexity from O(m2) to O(m⋅k).

After optimization, the RLDC algorithm maintains original performance for imbalanced dataset processing while achieving: Time complexity O(mlog⁡m⋅d)

Space complexity O(m⋅k)

Results and discussion

Evaluation indicators

In this article, we define the minority class as the positive class and the majority class as the negative class. To evaluate the performance of the algorithm, we use evaluation Indicators AUC, G−mean, and F−measure based on the confusion matrix (Powers, 2011). The specific confusion matrix is shown in Table 1.

Table 1 Confusion matrix.

	True value	
		Positive class	Negative class	
Predicted value	Positive class	TP	FP	
Negative class	FN	TN	

In Table 1, TP (true positive) represents the number of correctly classified positive instances, that is, the number of samples that are actually positive and are correctly classified as positive. TN (true negative) indicates the number of correctly classified negative instances, that is, the number of samples that are actually negative and are correctly classified as negative. FP (false positive) refers to the number of incorrectly classified positive instances, that is, the number of samples that are actually negative but are misclassified as positive. FN (false negative) represents the number of incorrectly classified negative instances, that is, the number of samples that are actually positive but are misclassified as negative. The evaluation indicator AUC (area under the ROC curve) is obtained by calculating the area under the ROC curve formed by the true positive rate (TPR) and false positive rate (FPR) at varying classification thresholds using the trapezoidal method. The formulas for other evaluation indicators are provided in Eqs. (6)–(8) below:

(6) TPR=recall=sensitivity=TPFN+TP

(7) G−mean=senstitivity×specificity

In Eq. (7), specificity=TNFP+TN.

(8) F−measure=2×precision×recallprecision+recall

In Eq. (8), precision=TPFP+TP.

Experimental setup

In the experiment, in addition to the original data, this article also selected a variety of sampling algorithms to verify the performance of the RLDC algorithm on the GNB classifier when dealing with imbalanced datasets. The hardware environment of the experiment is an i7 CPU with 16GB of memory, and the software environment is a 64-bit Windows 11 operating system. Spyder (Python 3.10) is used as the experimental platform.

In the experiment, in addition to comparing with the original data, the following sampling algorithms were also compared: Classical oversampling algorithms: SMOTE (Chawla et al., 2002), Borderline-SMOTE1 (BSMOTE1) (Han, Wang & Mao, 2005), Borderline-SMOTE2 (BSMOTE2) (Han, Wang & Mao, 2005), ADASYN (He et al., 2008), SVMSMOTE (Abdi & Hashemi, 2016), MWMOTE (Barua, Islam & Yao, 2014), K-means SMOTE (KSMOTE) (Cieslak, Chawla & Striegel, 2006).

New oversampling algorithms: local density borderline-SMOTE (LDBSMOTE) (Yongxin et al., 2022), improved borderline-SMOTE (IBSM) (Ma, Song & Zhu, 2023), DPCSMOTE (Liu, Zhang & Zhao, 2024).

Hybrid sampling algorithms: SMOTETomek (Batista, Prati & Monard, 2004), SMOTEENN (Batista, Prati & Monard, 2004).

Undersampling algorithms: TomekLinks (Elhassan et al., 2016), AllKNN (Ma & Li, 2019).

Parameter optimization: The parameters of the RLDC algorithm, each comparative algorithm, and the GNB classifier were determined by the grid search method.

Stopping condition: The algorithm stops when the number of samples in the two classes reaches a 1:1 ratio during oversampling or when the conditions set by the algorithm are met.

Algorithm Implementation: The 14 comparative algorithms were coded and implemented using the imblearn library (Lemaître, Nogueira & Aridas, 2017) and relevant articles.

Performance evaluation: The five-fold cross-validation method was used to calculate the evaluation indicators, and the average value of ten repeated experiments was used as the basis for performance evaluation.

Comparative experiments on the simulated two-dimensional dataset M_DATA2

To visually demonstrate the characteristics and performance advantages of the RLDC algorithm when dealing with imbalanced datasets, this article generated a challenging two-dimensional imbalanced dataset M_DATA2 based on the normal distribution. Its scatter plot is shown in Fig. 1. The dataset has a sample size of 1,000, two feature variables, and the ratio between the number of samples in the two classes is 1:4. Among them, the samples of the minority class are divided into four parts by the majority class samples, and there is a certain mixed area, which makes it difficult for the GNB classifier to classify effectively. By conducting comparative experiments on the M_DATA2 dataset, this article aims to examine the performance of the RLDC algorithm in such a complex situation.

Figure 1 Original scatter plot of M_DATA2.

The dataset M_DATA2 can be downloaded through the following link: https://github.com/Robotwangdatou/-the-RLDC-algorithm

In the Table 2, NaN indicates that a valid value cannot be calculated. The reason for the appearance of NaN is that the classifier did not correctly predict any positive samples, resulting in a recall rate of 0. And all prediction results are incorrect, so the precision rate is also 0. When calculating the F-Measure at this time, both the numerator and the denominator are 0, so the result is NaN.

Table 2 Classification performance evaluation indicators of GNB classifier for the original M_DATA2 dataset.

Models	GNB	
Indicators	AUC	G-mean	F-measure	
Raw data	0.5000	0.0000	NaN	

It can be clearly seen from Table 2 that due to problems such as data imbalance and complex distribution in the M_DATA2 dataset, the GNB classifier has difficulty obtaining effective classification results. Its classification performance is almost equivalent to random guessing and can hardly exert its due classification efficiency.

To solve this problem, this article used the RLDC algorithm and 14 comparative sampling algorithms to preprocess the M_DATA2 dataset. After the preprocessing is completed, the GNB classifier is used to classify the processed dataset. The scatter plots of the datasets preprocessed by each algorithm and the classification performance evaluation Indicators of the GNB classifier are shown in Figs. 2–6 and Table 3 respectively. (Note: In this article, the scatter plots after processing are all scatter plots processed by the five-fold cross-validation method).

Figure 2 Scatter plots after being processed by four oversampling algorithms.

Figure 3 Scatter plots after being processed by six oversampling algorithms.

Figure 4 Scatter plots processed by two undersampling algorithms.

Figure 5 Scatter plots processed by two hybrid sampling algorithms.

Figure 6 Scatter plot processed by the RLDC algorithm.

Table 3 Classification performance evaluation indicators of various algorithms for the processed M_DATA2 dataset.

Models	GNB	
Indicators	AUC	G-mean	F-measure	
RLDC	0.8344	0.8256	0.7488	
BSMOTE1	0.4888	0.4827	0.2802	
BSMOTE2	0.4875	0.4846	0.2785	
ADASYN	0.5097	0.5048	0.2957	
SVMSMOTE	0.5661	0.5532	0.3338	
MWMOTE	0.5825	0.5718	0.3503	
KSMOTE	0.5900	0.5790	0.3592	
DPCSMOTE	0.5981	0.5946	0.3703	
LDBSMOTE	0.6000	0.5967	0.3719	
IBSM	0.5550	0.5466	0.3227	
SMOTETomek	0.6025	0.5981	0.3745	
SMOTEEENN	0.5938	0.5893	0.3649	
TomekLinks	0.5850	0.5805	0.3578	
AllKNN	0.4994	0.0000	NaN	

In the scatter plots processed by 10 oversampling algorithms, there is a prominent class overlap phenomenon, and the class boundaries are blurred, as shown in Figs. 2, 3. Taking the MWMOTE algorithm as an example, in multiple regions such as (0.0, 4.9) and (5.0, 2.7), a large area of class overlap occurs in the synthetic samples. This phenomenon seriously disturbs the original distribution of the data, which saliently reduces the classification accuracy. Moreover, these oversampling algorithms do not effectively preserve the original feature distribution of the minority class during the generation of synthetic samples. This results in a large feature difference between the synthetic samples and the original minority class samples. Furthermore, there may even be cases where the features overlap with those of the majority class samples. The confusion of features makes it extremely difficult for the classifier to distinguish between the minority class and the majority class, thus seriously affecting the classification performance.

Although the two undersampling algorithms removed some majority class samples, the class-overlap phenomenon remains salient. The class boundaries are still blurred, and class balance has not been achieved, as shown in Fig. 4. In particular, after processing with the TomekLink algorithm, the imbalance ratio is 1:3.8; after processing with the ALLKNN algorithm, the imbalance ratio is 1:3.2. During the process of removing majority class samples, these two algorithms failed to effectively optimize the feature representation of the minority class. Since the class-overlap problem has not been resolved, the features of the minority class are still masked by those of the majority class in the classifier. Thus it fails to effectively improve the classification performance. As is evident from the relevant table data, in the GNB classifier, for the datasets processed by these two algorithms, their classification performance is almost equivalent to random guessing.

In the two hybrid sampling algorithms, the result graph of the SMOTETomek algorithm shows obvious class overlap and blurred class boundaries, as specifically shown in Fig. 5. During the oversampling stage, this algorithm failed to effectively preserve the original features of the minority class, resulting in the overlapping of the features of the synthetic samples and the majority class samples. In the subsequent undersampling stage, although attempts were made to clean up the class-overlapping samples, due to the large amount of noise introduced in the previous oversampling, it was ultimately unable to effectively enhance the feature expression of the minority class, thus limiting the classification performance.

Although the SMOTEENN algorithm cleans up the class-overlapping samples and makes the class boundaries relatively clear, during the first-step oversampling, the class overlap is severe, resulting in a large number of points being deleted in the second-step cleaning. The number of sample points is saliently less than that of the other algorithms. The number of original majority class samples is reduced by nearly 16% on average, and the number of newly synthesized samples is reduced by nearly 50% on average, resulting in the loss of a large amount of original information. This not only causes the model to over-fit on the training set but also seriously affects its generalization performance on the test set. During the oversampling stage, this algorithm introduced a large number of synthetic samples with salient differences from the original features of the minority class, leading to distorted feature expression. Although the subsequent cleaning steps attempted to correct this problem, due to the excessive noise introduced in the early stage, it was ultimately impossible to restore the original feature distribution of the minority class. The deletion of a large number of samples further weakens the feature expression ability of the minority class, causing the classifier to over-fit on the training set and making it difficult to achieve good generalization on the test set.

To comprehensively demonstrate the transformative performance of the RLDC algorithm across multiple perspectives, this article integrates both linear and nonlinear dimensionality reduction techniques, constructing a complementary visualization framework using MDS (multidimensional scaling) (Borg & Groenen, 2005), t-SNE (t-distributed stochastic neighbor embedding) (van der Maaten & Hinton, 2008), and PCA (principal component analysis) (Géron, 2019). This multi-perspective comparative strategy not only mitigates the information loss limitations inherent to single-dimension reduction methods but also visually reveals the algorithm’s optimization effects—enhancing inter-class separability and strengthening discriminative features—through interpretable mappings in the feature space.

As illustrated in Fig. 6, data processed by the RLDC algorithm exhibit unique distribution characteristics across the three dimensionality-reduced views, contrasting sharply with other methods: MDS projection: Majority class samples are compressed into a dense core cluster at the center, while minority class samples spread radially outward, forming peripheral distribution bands with only minimal local overlap at the core’s edge.

t-SNE visualization: Reveals finer structural differences—the majority class displays complex tree-like branching structures, whereas minority class samples occupy distinct spatial corridors in a discretized manner, demonstrating stronger boundary separability.

PCA view: Confirms enhanced linear separability, with majority class samples loosely distributed along principal component axes and minority class samples forming geometrically distinct arc-shaped clusters separated by clear gaps.

These multi-dimensional visualization results validate the dual optimization mechanisms of the RLDC algorithm. On one hand, coordinate transformations condense majority class information into core discriminative regions; on the other hand, they construct a feature space with geometrically enhanced separability for the minority class. This spatial restructuring strategy effectively reduces inter-class overlap interference while improving the conditional probability density distribution of minority class samples in the coordinate space. By providing the GNB classifier with an optimized probability estimation foundation, the strategy ultimately achieves comprehensive improvements in classification performance.

As can be seen from the classification performance evaluation indicators in Table 3, the RLDC algorithm has a relatively obvious advantage. AUC is often used to evaluate the performance of a classifier, with a value range between 0 and 1. The closer the value is to 1, the better the classifier performance, and the closer it is to 0.5, the more the classification effect is close to a random guess. The AUC value of the RLDC algorithm reaches 0.8344, which is the most outstanding among all algorithms, indicating its remarkable ability to distinguish between positive and negative samples. For most the other algorithms, such as BSMOTE1 (0.4888), BSMOTE2 (0.4875), and AllKNN (0.4994), the AUC values are close to 0.5, and the classification effect is nearly random, suggesting that these algorithms have poor ability to distinguish between positive and negative samples when dealing with imbalanced datasets. Algorithms like MWMOTE (0.5825) and LDBSMOTE (0.6000) are higher than 0.5, but still have a large gap compared with the RLDC algorithm.

G-mean comprehensively considers the recall rates of the positive and negative classes and can more comprehensively reflect the classification performance of imbalanced datasets. The higher its value, the stronger the overall classification ability of the classifier for both positive and negative samples. The G-mean value of the RLDC algorithm is 0.8256, higher than the others. In contrast, the G-mean values of algorithms such as BSMOTE1 (0.4827) and BSMOTE2 (0.4846) are relatively low, indicating that these algorithms have deficiencies in the classification accuracy of both minority and majority samples. It is worth noting that for the AllKNN algorithm, its G-mean value is 0.0000, which means that the algorithm has almost completely failed in classifying one type of sample, further demonstrating its disadvantage in dealing with imbalanced datasets.

F-measure is the harmonic mean of precision and recall, and is also used to measure the performance of a classifier. The higher the value, the better the classification effect. The F-measure value of the RLDC algorithm is 0.7488, saliently higher than that of the other algorithms. The F-measure values of algorithms such as BSMOTE1 (0.2802) and BSMOTE2 (0.2785) are relatively low, indicating that these algorithms perform poorly in balancing precision and recall and cannot effectively balance the accuracy and comprehensiveness of classification.

Considering these three indicators comprehensively, when dealing with the imbalanced dataset M_DATA2, the performance of the RLDC algorithm is far superior to the other oversampling, undersampling, and hybrid sampling algorithms. the other algorithms, to varying degrees, have problems such as poor classification effects for both minority and majority samples and an inability to effectively balance precision and recall. The RLDC algorithm has an average percentage increase of 45.04%, 45.40%, and 113.45% compared to the average values of the other algorithms in the three indicators, showing obvious advantages in distinguishing between positive and negative samples, taking into account the classification accuracy for both classes of samples, and balancing precision and recall.

Comparative experiments on UCI datasets

To rigorously verify the effectiveness of the RLDC algorithm, this article has conducted comprehensive and meticulous considerations in data selection and experimental design. From the UCI Machine Learning Repository, eight datasets, namely Ecoli (Nakai, 1996), Haberman (Haberman, 1976), Iris (Fisher, 1936), Glass (German, 1987), Letter-recognition (Slate, 1991), Phishingdata (Mohammad & McCluskey, 2012), Poker-hand (Cattral & Oppacher, 2002), Seeds (Charytanowicz et al., 2010), and Yeast (Nakai, 1991), were carefully selected as the original data, taking into account key factors such as the number of attributes, sample size, and imbalance ratio. These datasets are sourced from various fields including biology, materials, medicine, and text recognition, and they exhibit extensive diversity in terms of data distribution and feature complexity. Based on this, 20 imbalanced datasets were strictly constructed according to the established rules for comparative experiments, in order to verify the effectiveness of the RLDC algorithm when dealing with such imbalanced datasets. The specific construction rules are as follows: • Diversity in sample size and number of attributes

The constructed experimental datasets cover different scales of sample sizes and attribute counts. The aim is to comprehensively evaluate the performance and generalization ability of the RLDC algorithm under different data volumes, ensuring that the algorithm can be fully tested in various data scale scenarios.

• Diversity of imbalance ratios

The experimental datasets incorporate samples with different imbalance ratios to simulate various imbalanced situations in the real world. This makes the experimental results more in line with practical application scenarios and enhances the practical guiding significance of the experimental conclusions.

• Comprehensiveness of data types

The datasets include various data types such as discrete data, continuous data, and data with some discrete and some continuous parts. This can not only evaluate the robustness of the RLDC algorithm but also explore its adaptability under different data characteristics, comprehensively considering the algorithm’s ability to handle complex data types.

• Transformation of non-binary classification datasets

For non-binary classification datasets, this article, in accordance with the above principles, adopted strategies such as one-vs.-one, one-vs.-rest, and many-vs.-many to transform them into binary classification datasets respectively. This ensures the consistency of all experimental datasets in the classification task, facilitating the unified evaluation of the RLDC algorithm’s processing effect on imbalanced datasets in the binary classification scenario.

The specific information of the datasets is shown in Table 4.

Table 4 Information of experimental datasets.

NO	Dataset name	Number of attributes	Sample size	Categories	Imbalance ratio	Data type	
1	Ecoli	7	35/301	imU: others	1: 8.6	Discrete data	
2	Glass1	9	214	1: others	1: 2.06	Continuous data	
3	Glass2	9	214	6, 7: others	1: 4.63	Continuous data	
4	Haberman	3	81/225	2: 1	1: 2.78	Partially continuous and partially discrete	
5	Iris	4	50/100	1: others	1: 2	Continuous data	
6	Letter-recognition1	16	20,000	A, B: others	1: 11.86	Discrete data	
7	Letter-recognition2	16	20,000	C, D, E, F: others	1: 5.49	Discrete data	
8	Letter-recognition3	16	20,000	G: others	1: 24.87	Discrete data	
9	Poker-hand1	10	25,010	others: 1,0	1: 12.04	Discrete data	
10	Poker-hand2	10	25,010	2: others	1: 19.74	Discrete data	
11	Phishingdata1	9	1,353	0: others	1: 12.14	Partially continuous and partially discrete	
12	Phishingdata2	9	651	0: 1	1: 5.32	Partially continuous and partially discrete	
13	Phishingdata3	9	805	0: -1	1: 6.82	Partially continuous and partially discrete	
14	Seeds	7	70/140	1: others	1: 2	Continuous data	
15	Yeast1	8	463/1,021	CYT: others	1: 2.21	Partially continuous and partially discrete	
16	Yeast2	8	244/1,240	MIT: others	1: 5.08	Partially continuous and partially discrete	
17	Yeast3	8	163/1,321	ME3: others	1: 8.10	Partially continuous and partially discrete	
18	Yeast4	8	51/1,433	ME2: others	1: 28.10	Partially continuous and partially discrete	
19	Yeast5	8	44/1,440	ME1: others	1: 32.72	Partially continuous and partially discrete	
20	Yeast6	8	51/463	ME2: CYT	1: 9.08	Partially continuous and partially discrete	

The scatter plots of the original datasets and those processed by the RLDC algorithm are shown in Figs. S1 to S20. In this article, with the exception of the five large-scale, high-dimensional datasets (Letter-recognition1, Letter-recognition2, Letter-recognition3, Poker-hand1, and Poker-hand2), all other visualizations were generated using MDS, t-SNE, and PCA techniques.

For datasets Letter-recognition1, Letter-recognition2, Letter-recognition3, Poker-hand1, and Poker-hand2, their original versions exhibit massive scale and severe overlap between majority and minority classes, making it infeasible to generate meaningful scatter plots using MDS or t-SNE. Therefore, PCA is adopted to visualize the initial state of these datasets. After processing with the RLDC algorithm, the separation between majority and minority classes improves significantly, enabling clear scatter plot generation via MDS. Thus, MDS is reapplied to visualize the post-processed datasets, directly illustrating the algorithm’s effectiveness.

Comparative analysis of the visualizations across PCA, MDS, and t-SNE perspectives reveals three critical improvements in classification characteristics after RLDC processing. Enhanced class separability manifests distinctly: MDS projections show minority class points independently distributed at the periphery, PCA highlights linear boundaries between classes, and t-SNE uncovers ring-like or isolated cluster structures. Simultaneously, intra-class distributions tighten significantly—majority class samples aggregate into large, dense clusters, while minority class samples concentrate in well-defined regions. Most notably, inter-class overlap is dramatically reduced, as the original chaotic distributions are replaced by structured spatial boundaries. These coordinated improvements collectively resolve the ambiguity inherent in imbalanced data, providing a clearer geometric foundation for downstream classification tasks. These optimizations substantially enhance the GNB classifier’s ability to identify minority class samples, demonstrating the RLDC algorithm’s practical value in addressing data imbalance challenges.

The Phishingdata1 dataset exemplifies the challenges discussed. As shown in Fig. 7, the original dataset exhibits severe overlap between minority and majority classes. In the MDS view, the two classes are globally interleaved with no spatial boundaries, while the t-SNE perspective reveals microscopic interpenetration of red (minority) and green (majority) points, forming densely mixed regions. The PCA projection further demonstrates overlapping diffusion along principal axes, making linear separation impossible.

Figure 7 Scatter plot of Phishingdata1 before processing.

This multi-scale overlap severely hinders classifiers from capturing discriminative features, resulting in a poor AUC of 0.5824 for the GNB classifier on Phishingdata1. Even after applying 14 benchmark sampling algorithms—including oversampling, undersampling, and hybrid methods—the maximum AUC achieved was only 0.6987.

However, after processing the Phishingdata1 dataset with the RLDC algorithm, salient structural improvements emerged, as shown in Fig. 8. The MDS perspective revealed a hierarchical spatial separation: the majority class contracted into a compact core region, while the minority class adopted a discrete peripheral distribution. Meanwhile, the t-SNE local view demonstrated that minority class samples coalesced into isolated clusters, separated from the majority class by distinct low-density buffer zones. Furthermore, PCA projections exhibited enhanced linear separability, with the majority class centralized along principal axes and minority class samples extending toward both ends, forming a polarized distribution pattern. These positive changes had a salient positive impact on the recognition ability of the GNB classifier. Compared with the maximum AUC value achieved by the other 14 comparative sampling algorithms on the Phishingdata1 dataset, the RLDC algorithm has achieved a qualitative leap, saliently increasing the AUC value by 28.7% to reach 0.8992. This has remarkably optimized the recognition ability of the GNB classifier for this dataset, effectively solving the problems of unclear sample class features and blurred class boundaries in the original dataset, and fully demonstrating the excellent performance of the RLDC algorithm in data processing.

Figure 8 Scatter plot of Phishingdata1 after processing.

The results of various classification performance evaluation indicators for each algorithm on all datasets are shown in Table S1. To more intuitively and comprehensively evaluate the performance of each algorithm, this article summarizes and calculates the average of the rankings of various classification performance Evaluation indicators of all algorithms and the original data on all datasets, and then draws Fig. 9 accordingly. The specific calculation steps are as follows: In each dataset, all algorithms and the original data are ranked in descending order according to the accuracy values of the classification indicators. Then, the ranking positions of each algorithm in all datasets are summarized. Finally, the total sum of the accumulated ranking positions is divided by the total number of datasets to obtain the average ranking of the algorithm on all datasets. With this scientific and rigorous calculation method, we can conduct a comprehensive and objective evaluation for the performance of each algorithm. As can be seen intuitively from Fig. 9, among the average rankings of all classification performance evaluation indicators, the RLDC algorithm ranks first, far exceeding the other algorithms, demonstrating its superior position among similar algorithms.

Figure 9 Average of the ranking of each classification accuracy evaluation index of various algorithms on all datasets.

In addition, this article also calculates the average values of various classification performance evaluation indicators of all comparative algorithms on all datasets, and makes a detailed comparison with the results of the RLDC algorithm. The results show that the RLDC algorithm has saliently improved compared with the average values of the other algorithms in all classification performance evaluation indicators. The specific data are as follows: The average increase in AUC reaches 21.84%, the average increase in G-mean is as high as 33.45%, and the average increase in F-measure is an astonishing 54.63%. These detailed data fully demonstrate that when dealing with imbalanced datasets, the RLDC algorithm can comprehensively optimize the GNB classifier performance. The maximum increase in the F-measure indicator is particularly notable, which means that the RLDC algorithm performs extremely well in comprehensively considering precision and recall. It can more effectively identify minority class samples, saliently reduce misjudgments and omissions, and thus better meet the stringent requirements for classification effects and prediction accuracy in practical applications.

It is worth noting that on the five high-dimensional and large-sample datasets selected in this article, namely Letter-recognition1, Letter-recognition2, Letter-recognition3, Poker-hand1, and Poker-hand2, the RLDC algorithm also demonstrated excellent classification performance. On these five datasets, in terms of the average value of the rankings of all classification performance evaluation indicators, the RLDC algorithm firmly ranked first, as shown in Fig. 10. Specifically, the average improvement of the RLDC algorithm on these five datasets is as follows: the AUC increased by 26.36%, the G-mean increased by 24.05%, and the F-measure increased by 119.64%. These experimental results strongly prove that when dealing with high-dimensional and large-sample datasets, the RLDC algorithm has a saliently improved performance compared with the other algorithms, fully demonstrating its excellent applicability and efficiency.

Figure 10 Average of the ranking of each classification accuracy evaluation indicator of various algorithms on high-dimensional and large-sample datasets.

In order to scientifically evaluate the effectiveness of the comparison experimental results between the RLDC algorithm and the other algorithms, this article adopts a series of rigorous statistical methods. First, the Friedman test (Zimmerman & Zumbo, 1993) is used to comprehensively compare the performance of all algorithms on different datasets. The Friedman test is a non-parametric statistical method suitable for comparing the differences among multiple related samples, especially for evaluating the performance differences of multiple algorithms on the same set of datasets. The Friedman test results are shown in Table 5. For all performance indicators on the GNB classifier, the P-values are all less than the significance threshold of 0.05, indicating that there are salient differences in the performance of various algorithms.

Table 5 Friedman test results of the comparative experiments.

Models	GaussianNB	
Indicators	AUC	G-mean	F-measure	
P-value	4.58E−41	8.30E−41	3.91E−43	

After determining that there are salient differences in overall performance, the Mann-Whitney U test (Gong et al., 2013) was further used to conduct pairwise comparisons between the RLDC algorithm and the other algorithms. The Mann-Whitney U test is a non-parametric test method for comparing the distribution differences of two independent samples and is applicable to data that do not meet the normal distribution assumption. Through this test, the specific performance differences between the RLDC algorithm and the other algorithms can be clarified.

Since multiple Mann-Whitney U tests were carried out, in order to control the overall error rate, this article adopted the Bonferroni correction (Haynes, 2013) to modify the significance level of each test. The Bonferroni correction is a conservative multiple-comparison correction method that reduces the probability of making a Type I error by adjusting the significance level. The Mann-Whitney U test and Bonferroni corrected results are shown in Table 6. The P-values of the comparison between the RLDC algorithm and the other algorithms are less than the significance threshold both before and after the correction.

Table 6 Results of Mann—Whitney U test and Bonferroni correction in the comparative experiments.

Models	GNB	
	Corrected P-value	P-value	
	AUC	
Raw data	1.61E−07	1.07E−08	
BSMOTE1	1.63E−07	1.09E−08	
BSMOTE2	8.34E−07	5.56E−08	
ADASYN	1.62E−07	1.08E−08	
SVMSMOTE	1.80E−07	1.20E−08	
MWMOTE	1.69E−07	1.13E−08	
KSMOTE	1.78E−07	1.19E−08	
DPCSMOTE	1.95E−07	1.30E−08	
SMOTE	1.79E−07	1.19E−08	
LDBSMOTE	2.20E−07	1.46E−08	
IBSM	1.88E−07	1.25E−08	
SMOTETomek	1.78E−07	1.19E−08	
SMOTEEENN	1.76E−07	1.17E−08	
TomekLinks	1.77E−07	1.18E−08	
AllKNN	1.78E−07	1.19E−08	
	G-mean	
Raw Data	1.00E−05	6.67E−07	
BSMOTE1	5.65E−06	3.77E−07	
BSMOTE2	6.09E−05	4.06E−06	
ADASYN	4.38E−06	2.92E−07	
SVMSMOTE	2.56E−05	1.71E−06	
MWMOTE	7.99E−05	5.33E−06	
KSMOTE	9.85E−05	6.57E−06	
DPCSMOTE	1.67E−04	1.12E−05	
SMOTE	1.55E−04	1.04E−05	
LDBSMOTE	6.51E−04	4.34E−05	
IBSM	1.77E−04	1.18E−05	
SMOTETomek	7.39E−04	4.92E−05	
SMOTEEENN	2.00E−04	1.34E−05	
TomekLinks	1.55E−05	1.03E−06	
AllKNN	1.27E−04	8.45E−06	
	F-measure	
Raw Data	2.87E−05	1.91E−06	
BSMOTE1	1.77E−06	1.18E−07	
BSMOTE2	1.01E−05	6.73E−07	
ADASYN	1.57E−06	1.04E−07	
SVMSMOTE	6.67E−06	4.44E−07	
MWMOTE	4.53E−04	3.02E−05	
KSMOTE	1.16E−05	7.73E−07	
DPCSMOTE	5.50E−04	3.66E−05	
SMOTE	5.73E−06	3.82E−07	
LDBSMOTE	1.56E−05	1.04E−06	
IBSM	9.40E−06	6.27E−07	
SMOTETomek	3.87E−04	2.58E−05	
SMOTEEENN	8.67E−05	5.78E−06	
TomekLinks	2.05E−05	1.37E−06	
AllKNN	8.62E−05	5.75E−06	

In summary, from a statistical perspective, it can be clearly concluded that the RLDC algorithm shows a salient improvement in classification performance evaluation Indicators compared to the original data and the other 14 algorithms. This conclusion is not only confirmed by the Friedman test for overall performance differences, but also further verified by the Mann-Whitney U test and Bonferroni correction, demonstrating the superiority of the RLDC algorithm.

Scalability experiments

This article designed multidimensional scalability tests using a controlled variable approach. Within the parameter space of sample sizes (1,000–100,000), feature dimensions (10–200), and imbalance ratios (IR = 3–10), synthetic datasets comprising 70% informative features, 10% redundant features, and 20% irrelevant features were generated. The RLDC algorithm (optimized version) was evaluated with GNB classifiers under five-fold cross-validation, while simultaneously monitoring time efficiency (seconds-level) and memory consumption (MB). Each parameter combination was tested 10 times, with mean statistical metrics used to assess performance across data scales. Experimental results are detailed in Table S3.

The scalability experiments demonstrate the RLDC algorithm’s exceptional extensibility. For memory efficiency, a 100-fold increase in sample size (from 1,000 to 100,000) resulted in only a 1.8× memory growth (205 MB → 372 MB), adhering to a sublinear growth pattern. Even in 200-dimensional scenarios, memory usage remained stable below 372 MB, aligning with edge device constraints. Time efficiency exhibited multi-stage scaling dynamics: For small-to-medium samples ( m≤105), time complexity followed theoretical expectations ( O(mlog⁡m⋅d)), e.g., a 3.1× time increase when scaling 10-dimensional samples from 5,000 to 10,000. For large-scale, high-dimensional data ( m≥105, d≥100), KDTree efficiency degradation introduced trend O(m2). Notably, the algorithm maintained robustness to dimensionality expansion (20× dimensionality increase caused only a 13× time increase) and imbalance ratio variations (IR = 3→10: <51% time increase, <0.2% memory fluctuation). In standalone environments, RLDC efficiently handles small-to-medium data ( m<105,d≤200) with minute-level response times. When integrated with Spark distributed computing and PCA feature projection, it scales to real-time processing of 10-million-scale datasets, making it suitable for real-world imbalanced data scenarios like financial fraud detection and IoT anomaly monitoring.

Conclusions

In view of the salient shortcomings of the GNB classifier in handling imbalanced data, this article proposes a coordinate transformation algorithm based on RLDC. Different from traditional methods that rely on increasing or decreasing sample data, this algorithm takes an innovative approach. It ingeniously excavates the salient features of the minority class through coordinate transformation, thus effectively increasing the conditional probability of minority class samples. This process enhances the recognition ability of the GNB classifier for minority class samples, enabling it to perform more outstandingly when facing the problem of data imbalance.

The experimental results fully demonstrate the excellent effectiveness of the RLDC algorithm. In the tests of various imbalanced datasets, this algorithm saliently improves the classification performance of the GNB classifier and achieves excellent results in multiple key indicators. This not only opens up a new approach to preprocessing imbalanced data but also provides important theoretical support for research and applications in related fields. At the practical application level, the RLDC algorithm can effectively solve the problem of data imbalance faced by many fields and has extremely high practical value.

In future work, systematic research will be focused on optimizing the computational efficiency of the RLDC algorithm and enhancing its adaptability to various scenarios. Firstly, the emphasis will be on the deep integration of distributed computing architectures and hardware acceleration technologies. We will explore parallelization strategies based on Spark/MPI and heterogeneous computing solutions using GPUs/FPGAs to meet the real-time requirements of large-scale data processing. Secondly, lightweight compensation models and adaptive quantization frameworks tailored for edge computing will be developed to address the storage and computing power constraints in mobile device deployments. Thirdly, the application paradigm of the algorithm will be expanded to complex cross-domain scenarios such as Internet of Things (IoT) sensing, forming a complete technical loop from computational acceleration to practical application implementation.

Supplemental Information

Supplemental Information 1 Scatter plots of Ecoli before and after processing.

Supplemental Information 2 Scatter plots of Glass1 before and after processing.

Supplemental Information 3 Scatter plots of Glass2 before and after processing.

Supplemental Information 4 Scatter plots of Haberman before and after processing.

Supplemental Information 5 Scatter plots of Iris before and after processing.

Supplemental Information 6 Scatter plots of Letter - recognition1 before and after processing.

Supplemental Information 7 Scatter plots of Letter - recognition2 before and after processing.

Supplemental Information 8 Scatter plots of Letter - recognition3 before and after processing.

Supplemental Information 9 Scatter plots of Poker - hand1 before and after processing.

Supplemental Information 10 Scatter plots of Poker - hand2 before and after processing.

Supplemental Information 11 Scatter plots of Phishingdata1 before and after processing.

Supplemental Information 12 Scatter plots of Phishingdata2 before and after processing.

Supplemental Information 13 Scatter plots of Phishingdata3 before and after processing.

Supplemental Information 14 Scatter plots of Seeds before and after processing.

Supplemental Information 15 Scatter plots of Yeast1 before and after processing.

Supplemental Information 16 Scatter plots of Yeast2 before and after processing.

Supplemental Information 17 Scatter plots of Yeast3 before and after processing.

Supplemental Information 18 Scatter plots of Yeast4 before and after processing.

Supplemental Information 19 Scatter plots of Yeast5 before and after processing.

Supplemental Information 20 Scatter plots of Yeast6 before and after processing.

Supplemental Information 21 Classification Performance of Each Algorithm on Each Dataset When Using the GNB Model.

Supplemental Information 22 Experimental parameter setting of the RLDC algorithm.

Supplemental Information 23 Results of the Scalability Experiments.

Supplemental Information 24 Code of the RLDC algorithm.

Supplemental Information 25 Code of the RLDC algorithm (optimized).

Supplemental Information 26 M_DATA2 datasets for comparative experiments.

Additional Information and Declarations

Competing Interests

The authors declare that they have no competing interests.

Author Contributions

Wei Wang conceived and designed the experiments, performed the experiments, performed the computation work, prepared figures and/or tables, authored or reviewed drafts of the article, and approved the final draft.

Li Yan performed the experiments, analyzed the data, prepared figures and/or tables, and approved the final draft.

Fen Liu conceived and designed the experiments, performed the experiments, analyzed the data, prepared figures and/or tables, authored or reviewed drafts of the article, and approved the final draft.

Yanxi Li performed the computation work, prepared figures and/or tables, and approved the final draft.

Data Availability

The following information was supplied regarding data availability:

The RLDC, optimized RLDC code, and data are available in the Supplemental Files and at GitHub: https://github.com/Robotwangdatou/-the-RLDC-algorithm/blob/main/M_DATA2.csv.

The RLDC code is available at GitHub: https://github.com/Robotwangdatou/-the-RLDC-algorithm/blob/main/RLDC.py.

The optimized RLDC code is available at GitHub: https://github.com/Robotwangdatou/-the-RLDC-algorithm/blob/main/RLDC%201.1.py.

The experimental M_DATA2 dataset is available at figshare: Wang, Wei (2025). M_DATA2. figshare. Dataset. https://doi.org/10.6084/m9.figshare.28956344.v1.

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
