# Peer review of "Improving Gaussian Naive Bayes classification on imbalanced data through coordinate-based minority feature mining"

_PeerJ Computer Science, doi:10.7717/peerj-cs.3003_

## Round 0.1 · original submission · Major Revisions

Reviewer 1 ·

Basic reporting

This paper addresses the issue of enhancing the performance of Gaussian Naive Bayes (GNB) classifiers on imbalanced datasets, clearly demonstrating significant applicability and research value. The research problem is well-defined, and the authors thoroughly describe the impact and underlying causes of data imbalance on GNB performance.

However, the manuscript still requires improvement in several areas:

Although the paper briefly mentions the quadratic complexity of the proposed algorithm, there is insufficient discussion regarding its practical runtime efficiency, memory usage, and scalability, particularly for large-scale datasets. The authors are advised to supplement this analysis with relevant experiments and validation.

While scatter plots of datasets before and after transformation are provided, the visualizations for high-dimensional data rely solely on dimensionality reduction methods (such as MDS and PCA). These reduction techniques may obscure the direct interpretation of feature transformation effects. It is suggested that additional visualization approaches or a comparative discussion of the effects of different dimensionality reduction methods be included.

The future work section only briefly mentions exploring multi-class problems. The authors should expand upon this by specifying more detailed directions, such as considering feature selection, compatibility with other classifiers, algorithm optimization strategies, or methods for enhancing computational efficiency.

Overall, this paper presents a clearly defined topic, novel methodology, rigorous experimental design, and significant results, demonstrating both theoretical and practical contributions. Considering the above points, I recommend that this manuscript be accepted for publication after appropriate revisions.

Experimental design

.

Validity of the findings

This paper addresses the issue of enhancing the performance of Gaussian Naive Bayes (GNB) classifiers on imbalanced datasets, clearly demonstrating significant applicability and research value. The research problem is well-defined, and the authors thoroughly describe the impact and underlying causes of data imbalance on GNB performance.

Reviewer 2 ·

Basic reporting

Paper introduces a new algorithm for naïve bayes classifier that consider local densities using local relative density changes (RLDC) that reveal dense minority class areas.Proposed method is compared to various resampling algorithms and results show improvement.
Decision: Revision. I believe the paper investigates an interesting approach and can be published upon some revisions. My comments are given below.

1- In some cases, there is no spacing after period in abstract.
2- The word “salient” is used many times without clear explanation. Salient is an abstract term. Authors should explain clearly what they mean by it. Do you mean informative? Significant? Is salient feature a specific form of informative feature? Or is it actually informative/significant feature? Do salient features have specific properties that we should know? If so, what are they? For example, the term “salient features of minority classes” is used multiple times in the paper. Are they not salient enough for majority but salient for minority? If not, why not just say “salient features”? If it is salient just for minority class, then what does that mean? How is that possible? If a feature separates minority and majority, then it is informative for both. Not only for one.
3- Line 197-201, this is not about imbalance but general class separation. Of course, classification performance may improve but not because something has been done about imbalance.
4- Line 204-206, again, explained proposed approach is not about imbalance. I believe, the focus is on local univariate density estimation that reveals minority in this paper, not “salient” features.
5- Line 215-216, “In an imbalanced dataset, the changes in the local density of minority - class samples are often 216 more complex and their features are more prominent compared to majority - class samples.” “-“ looks like it is used as parenthesis. “minority – class” should be “minority-class”. Otherwise, it means something different. All similar cases should be corrected in the paper.
6- Also line 215-216, I am not sure why minority class is more complex and prominent. It is less informative because of smaller size. Do you mean the estimation of distribution is more complex? Why prominent at all?

7- In line 222, what do you mean by stable? Do you mean that majority class pattern is apparent?
8- Another local density based approach, locally weighted learning, is not mentioned in the paper. It is also adapted to Naïve Bayes multiple times. Examples:

• Frank, E., Hall, M., & Pfahringer, B. (2012). Locally weighted naive bayes. arXiv preprint arXiv:1212.2487.
• Jiang, L., Cai, Z., Zhang, H., & Wang, D. (2013). Naive Bayes text classifiers: a locally weighted learning approach. Journal of Experimental & Theoretical Artificial Intelligence, 25(2), 273-286.
• Wu, J., Zhang, G., Ren, Y., Zhang, X., & Yang, Q. (2017). Weighted local naive Bayes link prediction. Journal of Information Processing Systems, 13(4), 914-927.
• Sağlam, F., & Cengiz, M. A. (2024). Local resampling for locally weighted Naïve Bayes in imbalanced data. Computing, 106(1), 185-200.
After reading references, it seems Sağlam and Cengiz (2024) is mentioned. There is a typo in references for word “Sağlam”. All references should be checked.
Locally weighted learning is still not mentioned.
9- Line 365, trapezoidal or similar approximations should be used for AUC. This one is a inaccurate approximation.
10- Line 367, “×” instead of “*”.
11- Line 384, should not SMOTEEENN be SMOTEENN?
12- Line 369-370, “Pr” is not italic but “ecision” is italic. The same is for Recall. Should be corrected.
13- Line 699 and 747, typo in the author name.

Experimental design

no comment

Validity of the findings

no comment

Additional comments

no comment

---

## Round 0.2 · accepted · Accept

The manuscript has been significantly improved in clarity, technical explanation, and experimental validation. The authors expanded the algorithmic complexity analysis, provided additional visualizations using diverse dimensionality reduction techniques, clarified key terminology, and strengthened the discussion around the method’s applicability to imbalanced data.

Reviewer 2 ·

Basic reporting

Comment 1: In some cases, there is no spacing after a period in the abstract.

Response: Accepted. All the cases where there was a lack of spacing after the full stops in the abstract have been corrected.

Reviewer comment: Accepted.
* * *
Comment 2: The word “salient” is used many times without a clear explanation. Salient is an abstract term. Authors should explain clearly what they mean by it. Do you mean informative? Significant? Is the salient feature a specific form of informative feature? Or is it actually an informative/significant feature? Do salient features have specific properties that we should know? If so, what are they? For example, the term “salient features of minority classes” is used multiple times in the paper. Are they not salient enough for the majority but salient for the minority? If not, why not just say “salient features”? If it is salient just for the minority class, then what does that mean? How is that possible? If a feature separates the minority and majority, then it is informative for both. Not only for one.
Response: Accepted. A detailed explanation of the salient features of the minority class has been added to Section 3.2 of the paper to better interpret the relevant concepts.
Revision:
In real-world imbalanced data classification scenarios, "salient features of minority classes" refer to those characteristics that exhibit high distinctiveness and strong discriminative power in differentiating minority class samples from majority class samples. These features demonstrate unique distribution patterns or higher occurrence frequencies in minority class samples, while appearing less frequently or exhibiting different patterns in majority class samples. From an information content perspective, the salient features of minority classes contain rich information that captures the essential attributes of minority samples. Compared to other features, they play a more critical role in identifying minority class samples and serve as key distinguishing criteria between minority and majority classes. Regarding their importance, these salient features are decisive for the accurate classification of minority class samples. When effectively identified and leveraged, they can significantly enhance a classification model's ability to recognize minority class samples in imbalanced data scenarios, thereby improving overall classification performance.
The uniqueness of "salient features of minority classes" lies in that, due to data distribution imbalance, the limited quantity of minority class samples causes these features to manifest more concentrated and prominent patterns in the minority class, clearly reflecting their distinctive characteristics. In contrast, within the majority class, these features become diluted or obscured by the abundance of samples and relatively uniform distribution of other characteristics, losing their discriminative prominence. For instance, in diabetes screening scenarios, salient features of the minority class (diabetic patients), such as fasting plasma glucose (FPG) and glycated hemoglobin (HbA1c) maintain stable distributions in the majority class (healthy individuals) within normal ranges (e.g., FPG < 5.6 mmol/L), while showing salient clinical threshold breaches (e.g., FPG ≥ 7.0 mmol/L) or chronic abnormality accumulation (e.g., HbA1c ≥ 6.5%) in diabetic patients. Statistical analysis and model interpretation techniques (e.g., SHAP values) can validate the strong discriminative power of these features. When integrated with appropriate algorithms, they substantially improve diabetic patient identification accuracy while aligning with medical diagnostic standards.

Reviewer comment: Accepted.
* * *
Comment 3: Line 197-201, this is not about imbalance but general class separation. Of course, classification performance may improve, but not because something has been done about the imbalance.

Response: Accepted. This part of the content in Section 3.2 has been revised to better illustrate the role of the salient features of the minority class in dealing with imbalanced data.

Revision:
The content before the modification
Moreover, in Bayes' theorem, the posterior probability is also related to the prior probability. In imbalanced datasets, the prior probability of the minority class is low. However, when the features of the minority class are salient enough to make it high, according to Bayes' formula, the posterior probability can still be high. This means that although the minority class has a small proportion in the samples (low prior probability), prominent features can compensate for this disadvantage, resulting in a high posterior probability for the minority class given the sample features. This synergistic effect of prior and posterior probabilities further emphasizes the importance of mining salient features of minority classes to improve the classification performance of GNB in imbalanced data.
The content after the modification
Meanwhile, in the Bayesian theorem, the posterior probability also relates to the prior probability. In imbalanced datasets, the minority class's low prior probability often leads to its neglect by Bayesian classifiers. However, when the minority class possesses salient features (e.g., "cross-region large-amount transfers" in fraudulent transactions or disease-specific symptoms in rare illnesses), their exceptionally high conditional probability can significantly amplify the posterior probability through the multiplicative effect of the Bayesian formula, thereby overriding the majority class's dominance in probability calculations. For instance, when a symptom exhibits a conditional probability of 0.8 in a rare disease but only 0.01 in common diseases, even with the rare disease's prior probability as low as 0.1%, its posterior probability can rise to 7.4%, triggering further diagnostic investigation. While this mechanism depends primarily on general class separation in balanced data, in imbalanced scenarios, the conditional probability must substantially exceed that of the majority class to offset prior probability disadvantages. This highlights how minority class salient features specifically address data imbalance challenges through targeted optimization, rather than relying solely on class separation improvements.

Reviewer comment: Accepted.
* * *
Comment 4: Line 204-206, again, explained proposed approach is not about imbalance. I believe the focus is on local univariate density estimation that reveals minorities in this paper, not “salient” features.

Response: Accepted. Thank you for your profound question. The content of this paragraph failed to fully explain the method proposed in this paper, and we have made revisions to this part. The RLDC algorithm proposed in this paper is not simply aimed at discovering the minority class. Instead, it is a mining strategy proposed to address the problem that the salient features of the minority class are masked in the scenario of imbalanced data. In terms of the characteristics of local relative density changes in the scenario of imbalanced data, the minority class is more complex than the majority class. Therefore, this belongs to the salient features of the minority class, but this aspect is hidden in the existing feature space, and it is difficult to play a role in the Gaussian Naive Bayes (GNB) classifier. Therefore, this paper proposes the RLDC algorithm, which amplifies and highlights this feature in the new feature space through coordinate transformation, enhances the conditional probability of minority class samples in the GNB classifier, and thus effectively addresses the problem of imbalanced data and comprehensively improves the overall classification performance.

Revision:
The content before the modification
In summary, the close relationship between salient features of minority classes and conditional probabilities provides a key approach to improving the classification performance of GNB in imbalanced data. Therefore, this paper considers using coordinate transformation to mine salient features of minority classes, enabling them to play a greater role in the classification mechanism of GNB, thereby comprehensively improving overall classification performance.
The content after the modification
In summary, the strong correlation between minority class salient features and conditional probabilities provides critical insights for enhancing GNB's classification performance in imbalanced data. However, in such scenarios, the majority class's global distribution dominates the feature space, causing the minority class's salient features to become obscured. Therefore, the RLDC coordinate transformation algorithm proposed in this paper fundamentally addresses data imbalance by uncovering these masked salient features of minority class samples at the data distribution level. This enables them to play a more substantial role in GNB's classification mechanism, thereby directly tackling data imbalance issues and comprehensively improving overall classification performance.

Reviewer comment: Accepted.
* * *
Comment 5: Line 215-216, “In an imbalanced dataset, the changes in the local density of minority-class samples are often 216 more complex and their features are more prominent compared to majority-class samples.” “-“ looks like it is used as a parenthesis. “minority – class” should be “minority-class”. Otherwise, it means something different. All similar cases should be corrected in the paper.

Response: Accepted. We have corrected the incorrect usage of hyphens throughout the full text (such as "minority-class"). Thank you for your reminder.

Reviewer comment: Accepted.
* * *
Comment 6: Also, lines 215-216, I am not sure why the minority class is more complex and prominent. It is less informative because of its smaller size. Do you mean the estimation of distribution is more complex? Why prominent at all?

Response: Thank you for your insightful question. The minority class is more complex and prominent because the "complexity and prominence" here refer to the multimodality or abrupt change in the local relative density of the minority class, rather than the complexity of the sample quantity or the global distribution. Traditional methods assume that "the sample size determines the amount of information," but they ignore the decisive role of local information entropy in the discrimination of features. Although the number of samples in the minority class is small, the information entropy value of a single sample in terms of the characteristics of local relative density change may be much higher than that of a single sample in the majority class. As a result, the information density advantage of the minority class in this regard is sufficient to offset the disadvantage in terms of the global quantity. RLDC magnifies and highlights this characteristic, transforming the information advantage of the minority class in this regard into conditional probability weights, thereby solving the problem of data imbalance. This mechanism is not necessary in balanced data, but it has a good effect in imbalanced scenarios. We have revised and supplemented the relevant content in the article to better illustrate this point.

Revision:
The content before the modification
In an imbalanced dataset, the changes in the local density of minority class samples are often more complex, and their features are more prominent compared to majority-class samples. In contrast, the local density of majority-class samples is relatively stable, with simple and smooth changes in local density.
The content after the modification
To effectively capture the spatial variation in local class proportions, this paper introduces the concept of local relative density. Local relative density is defined as the ratio of the number of minority class samples to the total number of samples within a radius neighborhood centered at a given sample point. In imbalanced datasets, salient differences exist in the local relative density variation characteristics between minority and majority classes。 The majority class samples exhibit highly uniform and stable local relative density distributions, characterized by feature homogeneity and high information redundancy. This makes their numerical dominance ineffective in contributing meaningful information to classification boundaries. Minority class samples, however, display pronounced multimodal characteristics. Their local relative density fluctuates dramatically across the feature space, with abrupt transitions between sparse and dense regions forming a complex distribution pattern that carries high information entropy. Near classification boundaries, the density mutations of minority class samples often encode critical discriminative information.

Reviewer comment: Accepted. There are several typos in the new content. They should be corrected.
* * *
Comment 7: In line 222, what do you mean by stable? Do you mean that the majority class pattern is apparent?

Response: Thank you for your attention to the details of the paper and your questions. In this paper, when describing the "stability" of the local relative density of the samples in the majority class, it means that the change in the local relative density of the majority class is relatively stable and simple. This mainly stems from the adequacy of the sample size and the simplicity of the distribution pattern of the majority class: The number of samples in the majority class is large, and they follow the law of large numbers. They are smoothly distributed in the feature space with low variance (such as a Gaussian distribution), resulting in minimal fluctuations in the proportion of samples within a local area. At the same time, the features of the majority class have strong homogeneity (for example, normal transactions are concentrated in common numerical intervals), with a high degree of information redundancy, and the distribution pattern is far from the classification decision boundary, leading to a gradual and continuous change trend in its local relative density, without the drastic fluctuations caused by the sample sparsity and distribution complexity (multimodality, abrupt density changes) of the minority class. This stability not only provides a reliable learning foundation for the model but also highlights the necessity of mining the salient features of the minority class in scenarios of data imbalance.
We have supplemented relevant content in the paper to further illustrate the characteristic of the "stable" change in the local relative density of the samples in the majority class, enhancing the comprehensibility of the concept. Thank you again for your valuable comments, which will help us to improve the details of the paper.

Revision:
The content before the modification
The main reasons are as follows:
 Unevenness of Data Distribution
In an imbalanced dataset, there is a large difference in the amount of data between the minority class and the majority class. Due to the sufficient amount of data in the majority class, its distribution pattern is relatively stable. For example, in a bank transaction dataset containing a large number of normal transactions and a small number of fraudulent transactions, the number of normal transactions (majority class) is large, and their feature distribution may be relatively concentrated around some common patterns. However, for fraudulent transactions (minority class), because of the small number, the appearance of each sample may have a relatively large impact on their overall distribution.
From a statistical perspective, the number of samples in the majority class is large enough so that its distribution can be approximated by the central limit theorem, making the changes in its local density relatively smooth. While the number of minority-class samples is small, it is difficult to form a stable distribution pattern. Therefore, the changes in its local density are easily affected by new samples and become complex.
 Sparsity of Feature Space
Minority classes are usually sparse in the feature space. Take image classification as an example. Suppose we want to classify normal images and images containing rare objects. Normal images (majority class) may occupy a relatively continuous area in the feature space because they have many common features, such as relatively fixed distribution patterns of color and texture. However, images containing rare objects (minority class) are more scattered in the feature space.
When a new minority-class sample is added, due to its sparsity in the feature space, it may cause a large change in the local density in a local area. For example, in a high-dimensional feature space, a new minority-class sample may open up a new sub-area, causing the local density to suddenly increase in this new area. While for the majority class, due to its density in the feature space, this rarely happens, and the changes in local density are relatively simple.
 Overlapping and Confusion between Classes
There may be feature overlap between the minority class and the majority class. In an imbalanced medical diagnosis dataset, healthy samples (majority class) and samples with rare diseases (minority class) may have some similar physiological indicators. When trying to distinguish these two classes, the changes in the local density of the minority class will become complex due to this overlap.
The samples of the minority class may have different changes in local density near different class boundaries due to confusion with the majority class. For example, in medical imaging data for distinguishing benign and malignant tumors, the imaging features of malignant tumors (minority class) may be similar to those of benign tumors (majority class) in some cases. This leads to complex changes in the local density of malignant tumors when approaching the feature area of benign tumors. For the majority class, due to its dominant position and relatively clear boundaries, the changes in local density are relatively simple.

The content after the modification
These distinct local relative density behaviors arise from three fundamental factors:
 Differences in Sample Size and Statistical Stability
The majority class, with its large sample size, adheres to the law of large numbers, where local statistical estimates converge to the true distribution. For instance, normal transactions (majority class) consistently maintain a stable proportion of 90% ± 2% within the 100–500 transaction amount range, resulting in minimal fluctuations in local relative density even when new samples are added. In contrast, the sparsity of minority class samples leads to severe local statistical volatility. For example, fraudulent transactions (minority class) may account for only 0.1% of samples in the 5,000–10,000 transaction amount range, but within specific sub-ranges like 8,000–8,500, their local relative density can abruptly surge to 15% due to sporadic high-value transactions, forming sharp peaks in distribution.
 Distribution Patterns and Information Structure Divergence
Majority class distributions are typically simple (e.g., unimodal Gaussian) with strong feature homogeneity. For instance, normal user logins cluster around local IP addresses, exhibiting smooth majority proportion curves where local relative density gradually increases from the center to the periphery (3% → 5% → 10%). Minority classes, however, adopt complex multimodal distributions with salient feature heterogeneity. Fraudulent transactions may cluster in two distinct high-risk zones: Zone A (late-night microtransactions with 15% local relative density) and Zone B (cross-border macrotransactions with 20% local relative density), separated by low local relative density transition regions. This creates an alternating "sparse-dense" pattern.
 Classification Boundary and Information Entropy Contrast
The majority class samples predominantly lie far from classification boundaries, contributing minimal discriminative information due to low local entropy. In contrast, minority class samples cluster near decision boundaries, where high local entropy renders them critical for classification. For example, in academic performance evaluation, average students (the majority) dominate low-score regions with limited informational value, while top performers (minority) near high-score boundaries carry decisive classification signals.

Reviewer comment: Accepted.
* * *
Comment 8: Another local density-based approach, locally weighted learning, is not mentioned in the paper. It is also adapted to Naïve Bayes multiple times. Examples:
• Frank, E., Hall, M., & Pfahringer, B. (2012). Locally weighted naive bayes. arXiv preprint arXiv:1212.2487.
• Jiang, L., Cai, Z., Zhang, H., & Wang, D. (2013). Naive Bayes text classifiers: a locally weighted learning approach. Journal of Experimental & Theoretical Artificial Intelligence, 25(2), 273-286.
• Wu, J., Zhang, G., Ren, Y., Zhang, X., & Yang, Q. (2017). Weighted local naive Bayes link prediction. Journal of Information Processing Systems, 13(4), 914-927.
• Sağlam, F., & Cengiz, M. A. (2024). Local resampling for locally weighted Naïve Bayes in imbalanced data. Computing, 106(1), 185-200.
After reading references, it seems Sağlam and Cengiz (2024) are mentioned. There is a typo in the references for the word “Sağlam”. All references should be checked.
Locally weighted learning is still not mentioned.

Response: Accepted. We have added citations of local weighted learning methods in the paper (Frank et al., 2012; Jiang et al., 2013), and supplemented the discussion on the differences between these methods and the RLDC algorithm in Section 2.

Revision:
The supplementary part of Section 2
Compared with locally weighted learning (LWL) [2,19,20,21], which also focuses on local data features, the RLDC coordinate transformation method proposed in this paper demonstrates salient differences in core mechanisms and objectives: LWL is a dynamic non-parametric approach that performs local fitting through real-time calculation of weights for neighboring samples around query points. It is suitable for general regression or classification tasks, relying on distance relationships in the original feature space without altering data distribution, but exhibits insufficient sensitivity to data imbalance problems. In contrast, RLDC explicitly reconstructs the feature space through global coordinate transformation as a parametric method in the preprocessing stage, specifically designed to enhance classification performance on imbalanced data. While LWL incurs higher computational costs due to real-time weight calculations, RLDC achieves efficient classification during inference through distribution optimization in the training phase, with stronger interpretability (visualizable geometric transformations). This enables RLDC to more reliably supply effective feature information to GNB classifiers, thereby improving classification performance under imbalanced data conditions.
The part of the modification and addition of the references
[2]Sağlam, F., & Cengiz, M. A. (2024). Local resampling for locally weighted Naïve Bayes in imbalanced data. Computing, 106(1), 185-200.
[19][19]Frank, E., Hall, M., & Pfahringer, B. (2012). Locally weighted naive bayes. arXiv preprint arXiv:1212.2487.
[20]Jiang, L., Cai, Z., Zhang, H., & Wang, D. (2013). Naive Bayes text classifiers: a locally weighted learning approach. Journal of Experimental & Theoretical Artificial Intelligence, 25(2), 273-286.
[21]Wu, J., Zhang, G., Ren, Y., Zhang, X., & Yang, Q. (2017). Weighted local naive Bayes link prediction. Journal of Information Processing Systems, 13(4), 914-927.

Reviewer comment: Accepted.
* * *
Comment 9: Line 365, trapezoidal or similar approximations should be used for AUC. This one is an inaccurate approximation.

Response: Accepted, Accepted. The calculation of the Area Under the Curve (AUC) using the trapezoidal rule has been clearly stated in Section 4.3.

Revision:
The content before the modification
The corresponding evaluation Indicators are calculated using the formulas shown in Equations(6), (7), and (8):
(6)
The content after the modification
The evaluation indicator AUC (Area Under the ROC Curve) is obtained by calculating the area under the ROC curve formed by the True Positive Rate (TPR) and False Positive Rate (FPR) at varying classification thresholds using the trapezoidal method. The formulas for other evaluation indicators are provided in Equations (6), (7), and (8) below:
* * *
Comment 10: Line 367, “×” instead of “*”.
Response: Accepted, it has been revised. "×" has been used to replace "*".
Revision:
The content before the modification
(7)
The content after the modification
(7)

Reviewer comment: Accepted.
* * *
Comment 11: Line 384, should not SMOTEEENN be SMOTEENN?

Response: Accepted. The spelling error here has been corrected. All "SMOTEEENN" in the text have been changed to "SMOTEENN".

Reviewer comment: Accepted.
* * *
Comment 12: Line 369-370, “Pr” is not italic, but “ecision” is italic. The same is for Recall. Should be corrected.
Response: Accepted. The formatting error here has been corrected.
Revision:
The content before the modification
In Equation (6), and.
(7)
In Equation(7) , .
(8)
In Equation (8),.

The content after the modification
(6)
(7)
In Equation(7), .
(8)
In Equation (8),.


Reviewer comment: Accepted.
* * *
Comment 13: Line 699 and 747, typo in the author name.
Response: Accepted. The spelling error of the author's name here has been corrected, and other references have been checked.
Revision:
The content before the modification
[2] Sa ¬lam, F., & Cengiz, M. A. (2024). Local resampling for locally weighted NaÔve 700 Bayes in imbalanced data. Computing, 106(1), 185-200.
[19] Ç nar, A. (2024). Multi-Class Classification with the Gaussian Naive Bayes Algorithm. Journal of Data Applications, 0(2), 1-13.

The content after the modification
[2] Sağlam, F., & Cengiz, M. A. (2024). Local resampling for locally weighted Naïve Bayes in imbalanced data. Computing, 106(1), 185-200.
[22] Çınar, A. (2024). Multi-class classification with the Gaussian Naive Bayes algorithm. Journal of Data Applications, 0(2), 1–13.

Reviewer comment: Accepted.

Experimental design

.

Validity of the findings

.

Additional comments

There are some minimal typos in the new content. They should be corrected before publication.